# Investigation and Simulation of the Surface Contact Characteristics of Sinter-Joined Binder Jetting Components

Alexander Rütjes [1], Lukas Stahl [2], Michael Müller [2,*] and Frank Petzoldt [3]

1 Volkswagen AG Wolfsburg, Berliner Ring 2, 38438 Wolfsburg, Germany; alexander.ruetjes@volkswagen.de
2 Institute of Dynamics and Vibrations, Technische Universität Braunschweig, 38106 Braunschweig, Germany; l.stahl@tu-bs.de
3 Fraunhofer Institute for Manufacturing Technology and Advanced Materials IFAM, Powder Technology, Wiener Straße 12, 28359 Bremen, Germany; frank.petzoldt@ifam.fraunhofer.de
* Correspondence: mi.mueller@tu-bs.de

**Abstract:** Binder jetting holds great potential for revolutionizing conventional production processes for high-performance components. However, current applications face significant challenges regarding the depowdering and cleaning of complex internal geometries. A novel approach to these challenges is splitting the desired component into smaller, easy-to-clean parts and using sinter bonding to achieve the desired shape. This greatly reduces manual cleaning and preparation time during the production since sintering is required in any case. The sinter joint is currently significantly weaker than the rest of the part and may pose a risk of failure. This study focuses on the effects of different parameters that influence the joint strength and the contact surface between two parts. With different experimental setups, a variety of influences is identified and quantified: depowdering air pressure, component orientation in the build box, initial contact pressure between the green parts, and macroscopic component deformation. The experimental results are supported by a modified Boussinesq contact model. Combining experiments and simulations, it was found that the relative contact between two green parts varies between 5 and 26%, depending on the parameter set used. In this study, the authors introduce the idea of two-part manufacturing in metal binder jetting and subsequent joining of the components in the sintering process.

**Keywords:** metal binder jetting; sinter joining; sinter bonding; depowdering; surface roughness; contact surface area



## 1. Introduction

Binder jetting is a 3D printing process for plastics and metals, in which powder is applied in layers with a roller and then a binder is dispensed into the powder bed via a print head. The component is created completely enclosed in the powder bed, so that excess powder must be removed from the surfaces and from internal structures after printing. The current state of the art is to perform this powder removal manually in a time-consuming manner using a compressed air lance, brush and vacuum [1]. The process is presented in Figure 1. In order to remove unbound powder from voids such as channel structures, openings need to be included in the design for effective depowdering of the parts through which the unwanted powder can trickle away during the powder-removal process [2].

Because final component strength in metal binder jetting is not achieved until undergoing a subsequent sintering process, the unsintered components (also called green parts) are very fragile and can easily be damaged during handling [3,4]. A particularly high risk of damage to the components arises from the manual depowdering process, e.g., for components with fine external features or internal hollow structures. Depending on the complexity of such a structure, it is sometimes impossible to entirely remove the excess powder. Remaining powder in the channels that is not removed is bonded to the actual

part area in the sintering process, thus significantly altering the desired component shape and function [5,6].

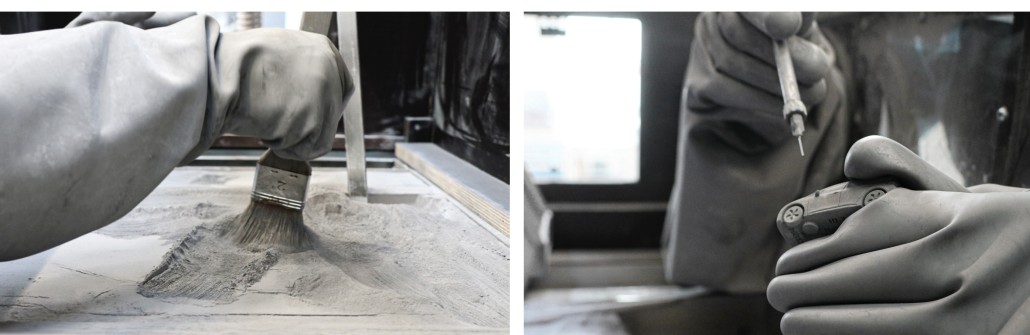

**Figure 1.** State-of-the-art methods for cleaning binder jetting green parts with brushes, vacuum and compressed air lance.

Figure 2 shows an example of this problem. The image shows a cut through a sintered part that features a curved cooling channel. During depowdering it is not possible to clean the entire channel, so unwanted residual powder remains in the middle section of the channel. Compared to the same component produced with the Laser Powder Bed Fusion process, this powder cannot be easily removed by using strong compressed air or tapping the component on a hard surface due to the fragility of the green part. Even light tapping, for example, would break off edges of the binder jetting green part, and if one were to use a compressed air lance to remove excess powder from channels, powder that is being swirled around in the jet of compressed air and hits the part may lead to surface erosion.

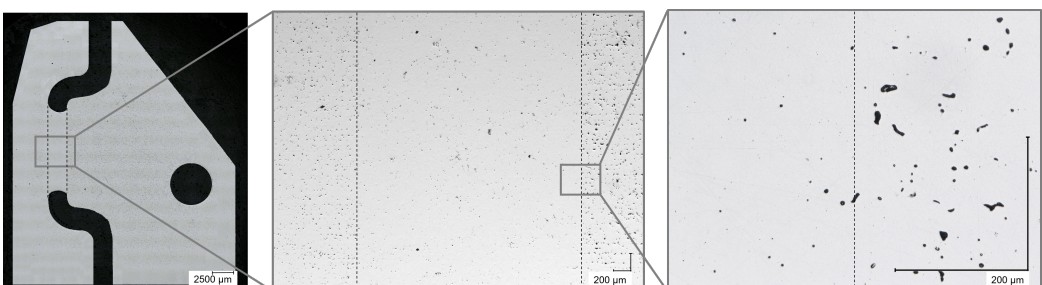

**Figure 2.** Clogged channel in a sintered binder jetting part.

The clogging powder actually sinters more densely than the actual component (compare middle and right image of Figure 2). This is caused by the applied binder. Just as in metal injection molding (MIM), a large amount of the binder is burned on a holding stage for debinding in the sintering furnace at approximately 500 °C, however, the binder component known as the backbone burns later on and slightly above the temperature at which the first sintering necks are formed between the particles. This is necessary because otherwise the powder particles would no longer be held together in the component geometry [7,8]. The authors assume that the presence of this backbone in this phase of the sintering process could have a significant negative influence on the compaction process, because, for example, distances between the particles are increased or the now evaporating binder interferes with the diffusion processes. A distance of 1 μm between two particles is sufficient to prevent the particles from sintering [9].

### 1.1. Aim of the Article

Design restrictions of metal binder jetting parts are currently limiting the industrial use of this production process, as parts with internal hollow structures of a certain complexity can not be fully depowdered. This paper addresses these limitations and investigates a possible modification to the production process, in which parts with complex channels

are divided into multiple components prior to printing. Subsequently, these components are rejoined during sintering by sinter bonding. Although it would be possible to use conventional joining processes, the suitability of the required sintering step for joining the components offers a large increase in efficiency as a densification and simultaneous joining process. The first and most practical step in the development of such a joining process is simply placing the components to be joined on top of each other. In order to establish a basic understanding for an economical future usage and to identify influencing factors on this new approach in binder-jetting technology, various variables were examined and a methodological approach for evaluating joint quality had to be developed.

The following studies present different variables influencing the sinter-joining process for split binder jetting green parts. In order to verify the general applicability of sintering as a joining technique for binder jetting, different influences on the quality of the joints are investigated based on relatively simple joining samples. The quality of the joint is evaluated on the basis of metallographic cross-sections. Since the quality of the joint depends to a large extent on the initial contact between the joining partners (see Section 1.2.1), the initial contact of the green parts is traced back from the sintered state to the green part state with the aid of a contact simulation, since there is no functioning preparation methodology which could be used to investigate the actual green part contact.

### 1.2. Prior Work on Sinter-Joining Processes for Powder Metallurgical Manufacturing

To date, there has been no published research on the sinter joining of binder-jetting components. In MIM, on the other hand, in which components are molded by injecting a mixture of metal powder and a thermoplastic binder into a tool, the limits of part sizes and part complexities that can be produced using tools were reached several years ago. Thus, sinter-joining methods have already been developed in MIM and are being used in economically relevant applications [10,11]. After the debinding step, MIM components have a density of approximately 60% [12], which should make them comparable to binder jetting green parts during sintering. In addition to joining processes that are MIM-specific and not transferable to other processes, such as simultaneous injection of different feedstocks into a mold [13–15] or overmolding of steel inserts [16], there are processes that could in principle also work in binder jetting. One possibility of MIM green part joining was demonstrated by Zhang et al. For this purpose, MIM green parts were joined using a powder interlayer consisting of a mixture of 10 wt% cooking oil and iron powder. The oil was added for its adhesive effect, since loose metal powder cannot be applied to complex structures with a homogeneous distribution and uniform layer thickness, according to Zhang et al. However, sintering temperatures above 1250 °C lead to degradation of the hydrocarbon compounds of the oil, resulting in undesirable carburization of the base material [17]. This can even lead to local melting of the material by exceeding the eutectic temperature. In addition to the aforementioned processes, brazing alloys have been developed that are placed in the joining zone to create a material-to-material bond. In this case, conventional brazing alloys cannot be used because the brazing material is drawn into the open-pored green part after reaching melting temperature due to capillary effects [9]. Therefore, special solders have been developed that react and solidify with the steel of the green part after reaching the melting temperature, so that capillary action is interrupted [8]. Disadvantages of brazing are that the joint is significantly weaker than the base material [18] and the gap height between the components must be precisely adjusted [19,20]. Further joining tests were carried out by Klimscha using microcomponents that were to sinter together by mere contact. It was shown that the success of the joining process strongly depends on the initial contact of the joining partners and that the initial contact depends on the shape of the joining surface, the accuracy of the manufacturing process and the arrangement of the green parts before the joining process. The shape deviations of the components to be joined correlated negatively with the achievable joint strength [21]. This was also shown by Pfeiffer et al. [22]. Roughness of surfaces to be joined is closely related to the initial contact of the surfaces. On a microscopic level, an increased roughness of the joining surfaces leads

only to punctual contacting, resulting in an insufficient joint. The roughness thus directly influences the strength of the joint [23]. Therefore, it is absolutely necessary to investigate the surface finish and the resulting joining quality when developing a joining method for a new manufacturing process.

### 1.2.1. Influence of the Initial Contact on the Diffusion Process

During quality control of sinter-bonded components, the bond strength is key, besides characteristics such as shape accuracy and surface properties. Several researchers have found that the most significant influence on a successful sinter joint of compatible materials is the initial contact of the joining surfaces of the green parts before the sintering process [24,25]. Jacobson and Humpston described the sinter-joining process as the joining process least tolerant of misfit of the joining surfaces [25]. Furthermore, this influence has been explicitly investigated by Klimscha [21], among others, for the effects on the achievable joint quality of MIM components. It is shown that the initial contact depends on the shape of the joining surface, the accuracy of the manufacturing process as well as the arrangement of the green parts before the joining process. Shape deviations of the components weaken the achievable joint strength. This is confirmed by Pfeiffer et al. [22] in the field of diffusion bonding. The large influence of the initial contact on joint quality is due to the contact of the particles required for material diffusion. This influence is further confirmed in literature on the application of MIM sinter joining, 2C-MIM (two-component MIM) and sinter brazing [19,20,25–28].

In contrast to MIM, properties of green parts in the binder jetting process differ not only in strength, but also primarily in the expression of surfaces, because the binder-jetting process, unlike MIM, is tool-less, and surfaces are created free in the powder bed instead of being an impression of a smooth tool surface [12,29]. Thus, although binder jetting allows for the flexible production of a wide variety of shapes, these have significantly rougher surfaces. The surface properties of binder jetting green parts are also subject to other factors, such as the orientation of the relevant surface in the printing process and the process control in the depowdering process [30]. Furthermore, a wide variety of phenomena can lead to a deviation from the components' nominal geometry in the form of warpage, such as negative pressure during the curing process, deviation in the stacking of the individual layers or binder bleeding [31,32].

## 2. Methods

### 2.1. Additive Manufacturing Process for Sample Preparation

The 3D printer used to fabricate the samples is a prototype system that is currently not openly available on the market and can therefore not be shown. The system has a build space of $430 \times 320 \times 200 \text{ mm}^3$. All printed samples are made with gas-atomized AISI 316L powder with a particle size distribution of $24\,\mu m$-d90 at layer heights of $70\,\mu m$. The printhead has a resolution of 1200 dpi, allowing for build rates of approximately $2000\,\text{cm}^3$ per hour to be achieved. The printer uses a solvent-based binder that has a latex backbone and is added to the green parts at about 1 wt%. After the printing process, the entire powder bed is heated to evaporate the solvent component of the binder and thus cure the binder. This process step is carried out under continuous vacuum generation by a vacuum pump, with which the solvent vapors are extracted downwards through the powder bed and through holes in the build platform. In the next step, the specimens are removed from the powder bed and depowdered manually.

### 2.2. Sintering of the Specimens

Sintering of the specimens (when required) is performed in a Nabertherm VHT40/16 cold wall retort furnace under 100% hydrogen atmosphere with a sintering temperature of 1380 °C for two hours on alumina setters. The furnace is a standard sintering furnace and is displayed in Figure 3.

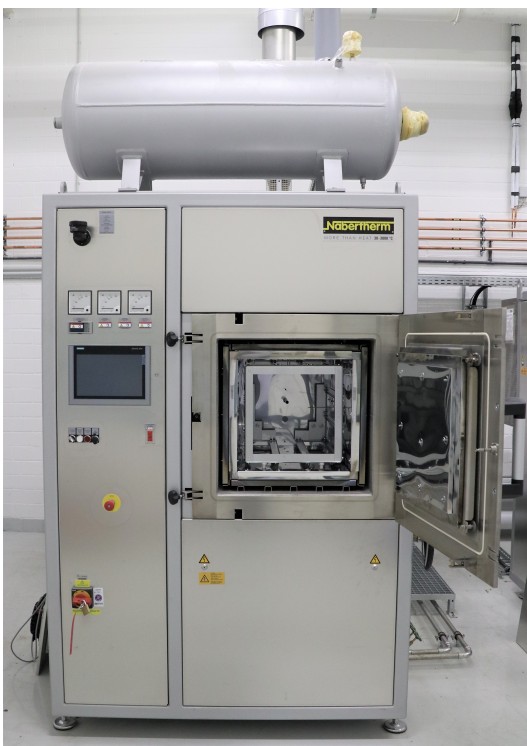

**Figure 3.** Nabertherm VHT40/16 cold wall retort furnace used to sinter the parts presented in this study.

### 2.3. Methodology for Evaluating Joint Quality

Various scientific studies have shown that the mechanic joining strength of two powder metallurgically produced components is largely determined by the quality of the joining zone. Defects within the joining zone such as pores strongly reduce the strength [18,33]. German [34] found that for sintered steel, an increase in density from 90% to 100% nearly doubles the tensile strength. Surface roughness leads to an increase in the distance between the joining surfaces and thus makes the joining process more difficult, so the surfaces should be as smooth as possible [21,25].

Evaluating the joint quality can be very challenging and is usually carried out on the basis of mechanical properties by means of tensile tests [16,21]. However, for this study, a new methodological approach is developed to investigate the contact ratio of the sintered joint. The basis for this is the metallographic preparation of the specimen shown in Figure 4.

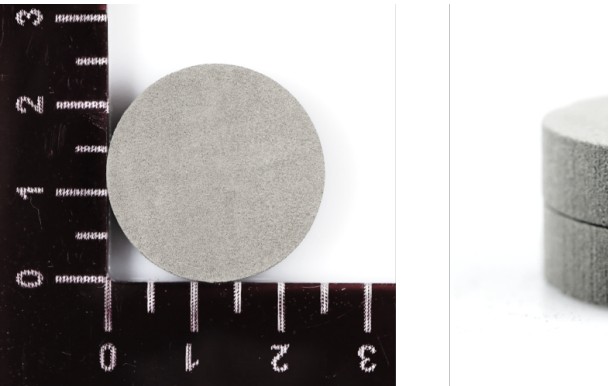
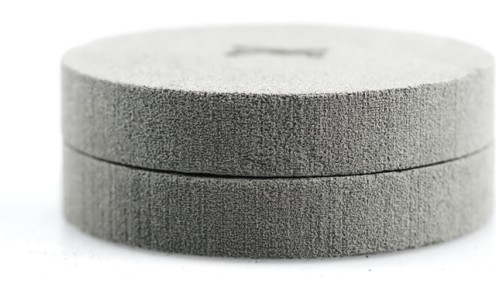

**Figure 4.** Size of the selected flat cylinder to determine the contact area after sintering (**left**) and stacked green parts before sintering (**right**) (scale in centimeters).

This method provides a closer insight into the joint itself and facilitates the analysis of the contact ratio of the joint. The contact ratio is described as the ratio of the area of both components in contact to the unconnected area, based on the total sample length. To experimentally investigate this ratio, the manufactured and sintered specimens are cut in half orthogonally to the joint plane and both sides are metallographically prepared so that the cutting cross-section is visible. Then, after grinding and polishing, a panoramic image of the contact zone is taken with a Keyence VHX-5000 microscope, which is a standard method in metallography. Using an automated color-contrast algorithm of the Keyence software, the remaining gap between the parts of the joining zone is colored green (compare Figure 5). The resulting image is fed into a developed Matlab script, which For this, the vertical pixel rows of the images are evaluated pixel-by-pixel for the occurrence of the coloring. If the script identifies green pixels, this area is classified as not connected, and vice versa. The ratio of connected to unconnected areas is then determined along the length of the contact area to be measured. The procedure is visualized in Figure 5. This methodology also allows for a determination of the average gap height by automatically counting the number of green pixels per vertical pixel row and calculating the average value over the entire sample length. This can then be converted into a micrometer measurement based on the magnification.

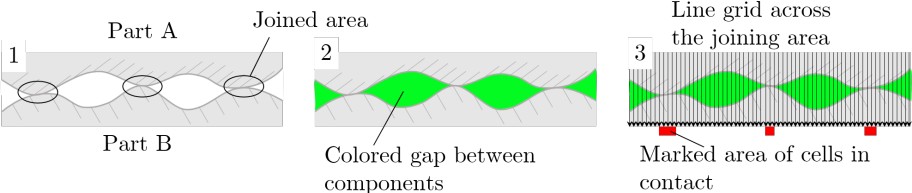

**Figure 5.** Procedure and principle functionality of the contact area analysis.

This procedure yields the contact ratio across the entire cross-section, but it only represents a one-dimensional example measurement of the entire contact area. In addition, it can only be applied to sintered parts, as green parts can not be metallographically analysed by this method since they would simply fall apart. All specimens are roughly cut in the middle, and due to the cylindrical specimen shape, it is not expected that significant deviation occurs over other cross-sections of the contact area. To transfer these experimental data from the sintered part to the green part contact and from the two-dimensional analysis of sections to the full contact zone, a simulation model is used (see Section 2.6).

### 2.4. Specimen and Experiments for Roughness Evaluation

Typically, part of the depowdering treatment is cleaning the green part with an air lance and compressed air. To investigate the influence of the air pressure on the surface characteristics, specially housed samples are produced and are shown in Figure 6. This special geometry is chosen so that a protective cage is created around the measuring surface to prevent it from being affected beforehand when the specimen is removed from the powder bed directly after printing. The authors are not aware of any such specimen in binder jetting to date to determine roughness without error.

Different air pressures at a single tip nozzle with an opening diameter of 0.8 mm are used to depowder the samples. The compressed air levels 2 bar, 3 bar, 4 bar as well as 5 bar are preset with a manometer. The process is carried out manually, so the depowdering time varies slightly within and between measurements of different settings of air pressure. Depowdering is always terminated immediately when the surface appears to be free of loose powder to minimize time differences. Although there are potential deviations between individual measurements as a result of this manual process, this procedure represents the current state of the art of powder removal in binder jetting. For each pressure level, five samples at two randomly selected areas are evaluated with the confocal microscope

MarSurf CM mobile from the manufacturer NanoFocus AG with regard to the roughness of the measuring surface.

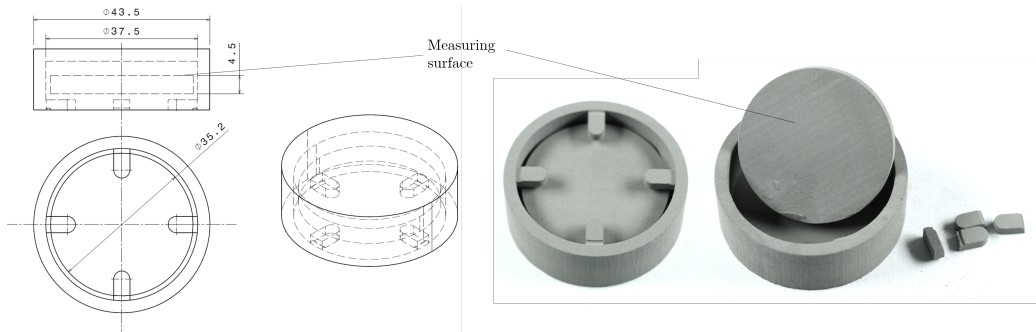

**Figure 6.** Specimen geometry used for experiments (dimensions in mm).

Investigation into the influence of print orientation on the surface roughness is also carried out using the specimen shown in Figure 6. The experiments are conducted separately from those previously described. To analyze the influence of print orientation on the process, the entire assembly must be placed in the data preparation prior to printing with respect to the orientation to be evaluated. In this study, the three main orientations (top-side, bottom-side and 90°-side surface) are considered. The depowdering pressure is set at 2 bar as a result of the evaluation of the aforementioned analysis (compare Section 3.1) to produce the smoothest surfaces possible. Three areas on the sample surface are randomly selected per sample and a squared area with an edge length of approximately 2.1 mm is analyzed by confocal microscopy. Overall, 10 samples are prepared and evaluated, therefore a total of 30 measurements per orientation are carried out.

### 2.5. Investigation into the Print Orientation on the Sinter Joint

The print orientation is analyzed by determining the contact ratio for sinter-bonded parts, as described in Section 2.3. For this, a smaller version of the specimen shown in Figure 6 (see Figure 9 left) is printed with different orientations (top-side, bottom-side and 90°-side surface). Then, two identically oriented samples are placed congruently on top of each other (see Figure 4 right). Six such assemblies are printed per orientation, sintered, sectioned, metallographically prepared and analyzed regarding contact ratio after sintering.

### 2.6. Model for Contact Simulations

To understand the three-dimensional contact behavior of the specimens during production, a simulation with a simple contact model is carried out. The model enhances the understanding of the complex interaction of the topography of the surfaces. The model is not used to predict the contact area of sintered green parts because there is no model available that accounts for the complex contact mechanics during the sintering process. The contact model is instead based on a Boussinesq approach [35]. The surface topographies used in the simulation are created with surface measurements from real, unsintered specimens. The specimens were measured with the mobile confocal microscope MarSurf CM mobile from the manufacturer NanoFocus AG, which is also used in this study to measure surface roughness.

Two different specimens were measured: the upwards- and sidewards-facing surfaces (compare Section 2.5). From these initial measurements, equal sized cutouts were used as simulation topographies. Figure 7 shows an overview of two the topographies, the left topography is from the upwards-facing specimen and the right topography is from the sidewards-facing specimen. The surfaces of the topographies are shown from the top view. The topography of the upwards-facing surface is relatively smooth (see colorbar and scale of z-axis) and individual powder particles are visible. The topography of the

sidewards-facing surface is rougher, and the individual, horizontally directed print layers are visible.

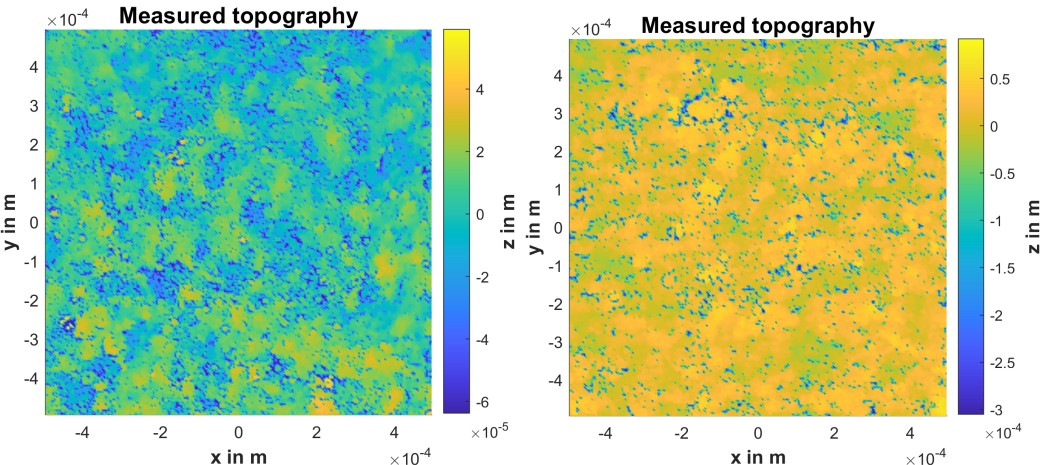

**Figure 7.** Measured topography used in the simulations for the (**left**) upwards- and (**right**) sidewards-facing surface.

For this model, the material is treated as homogeneous and solid. To account for the complex material behavior of the green part, the material parameters are modified and plastic deformation is introduced into the model so that the normal stress is limited. While this does not represent the complex contact mechanical behavior of the composite material or the nonlinear material behavior during sintering, it helps to understand the influence of the surface roughness and structure on the contact ratio and on the material deformation. The model gives insights into the contact geometry before sintering.

In the simulation, the material is modelled with two parameters—the Youngs Modulus and the elasticity limit. The former represents the elastic resistance against deformation and the latter represents the maximum pressure for the elastic deformation. Any further pressure will lead to plastic deformation. The simple contact model of this simulation assumes that the load-carrying ability is limited to this elasticity limit, after which the cell will continuously plastically deform. The parameters were derived from the experiments and were set to $3 \times 10^9$ Pa for the Youngs Modulus and to $2 \times 10^6$ Pa for the elasticity limit. The domain was discretized with $251 \times 251$ points.

During the simulation, an iterative Newton algorithm is used to find the balance between gap height and applied normal stress. A starting gap height (with its correlated penetration) is chosen and the resulting normal stress is calculated. If the stress thus calculated is lower than the target stress, the gap height is decreased and vice versa. Simple plastic deformation is modelled by limiting the normal stress that an individual contact cell can transmit. The contact cell experiences further plastic deformation, but does not transmit more stress than the threshold. The results show that a significant amount of cells are plastically deformed (and thus transmit the maximum normal stress). For the brittle green parts, this is a reasonable behavior, however the behavior is directly influenced by the initial set of parameters.

Figure 8 shows an overview of the surfaces before the start of the simulation. Two different cutouts from the same measurements are used for the upper and the lower topography. The deep incisions visible from the side are measuring points that could not be recorded with the confocal microscope because they were outside the measuring range or were shadowed.

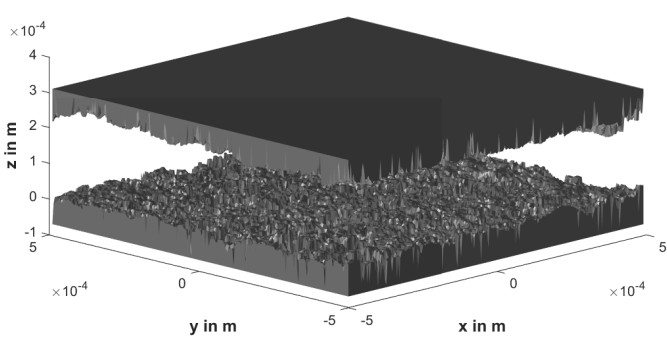

**Figure 8.** Upwards-facing topographies at start of simulation.

### 2.7. Influence of Assembly Force on the Sinter Joint

The assembly force is investigated as another variable on the quality of a joint after sintering. The assembly force is defined as the force applied onto the two green components during the placement process before sintering. Due to the low strength of the green part, it is expected that local asperity peaks will break away when the components are brought together, thus potentially increasing the contact ratio. For this purpose, the smaller version of the specimen which was already introduced in Section 2.5 is fabricated (see Figure 9). Then two such specimens, each with contacting top sides, are placed on top of each other and a microuss is used to manually apply a force of 0 N, 50 N, and 100 N. To prevent the risk of damaging the green parts, using a force higher than 100 N was abstained from in this study. In addition to the force that is being applied, the dead weight of the top part (10.05 g) is taken into account. After the desired force is reached, the force is released and only the dead weight remains. For each force level, three assemblies are created. The resulting assemblies are sintered, metallographically prepared, and the contact ratio is again evaluated.

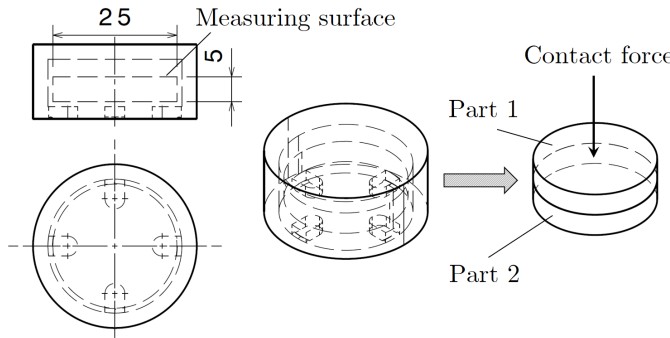

**Figure 9.** Specimen used to evaluate the influence of the contact pressure on the sintered joint (dimensions in mm).

### 2.8. Measuring of the Component Shape Deviation

Another important influence on the bond strength is macroscopic shape deviation of the components. Obviously the maximum amount of bonds between the two green parts can only be achieved with maximum planarity of the surfaces. Therefore, the build box of the printer was filled with flat cylindrical specimens with a diameter of 25 mm and a height of ca. 4.5 mm at the lowest build level (see Figure 10). To investigate the occurrences of shape deviations depending on the position in the build space, the position of the specimens inside the build box of the printer is tracked by imprinting numbers. After fabrication with the parameters described in Section 2.1 and manual depowdering, all samples are digitized with the VL-500 3D-scanner from the Keyence company. Three-dimensional scanning is broadly adopted to assure part quality in three-dimensional printing and other

manufacturing techniques. In the manufacturer's analysis software, the flatness of the topsides of the samples can easily be evaluated. The flatness is considered as the basis for the evaluation. This is the vertical height difference between two imaginary parallel planes on which the lowest and highest points of a component lie.

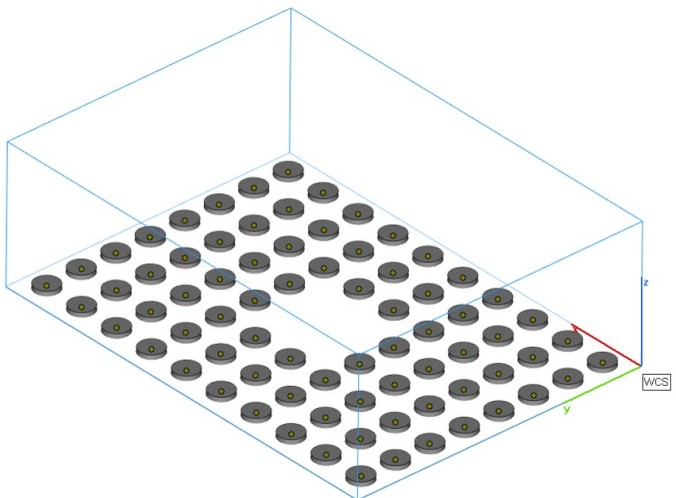

**Figure 10.** Print job layout for measuring the warpage characteristics across the build platform.

## 3. Results

In the following section, the results of the aforementioned methods will be displayed. First, the surface characteristics are shown. They depend on the air pressure used during depowdering as well as the orientation in the printing process. From these results, an air pressure is selected with which the smoothest surface properties can be obtained. All further tests are carried out with said air pressure. This is followed by an analysis of the variables influencing the quality of the sintered joint: orientation, assembly force and component shape deviation. The resulting contact after sintering is also traced back to the green part contact by the simulation.

### 3.1. Influence of Depowdering Air Pressure on the Surface Roughness

Figure 11 shows the results of this experiment. For the surface analysis, the roughness parameters for a plane ($Sa$, $Sq$) are used to ensure that the profile evaluation is performed correctly. When using 2D roughness values, it must be ensured that the parts are examined in the build-up direction, since an evaluation parallel to the stacked layers will result in an underestimation of the actual roughness.

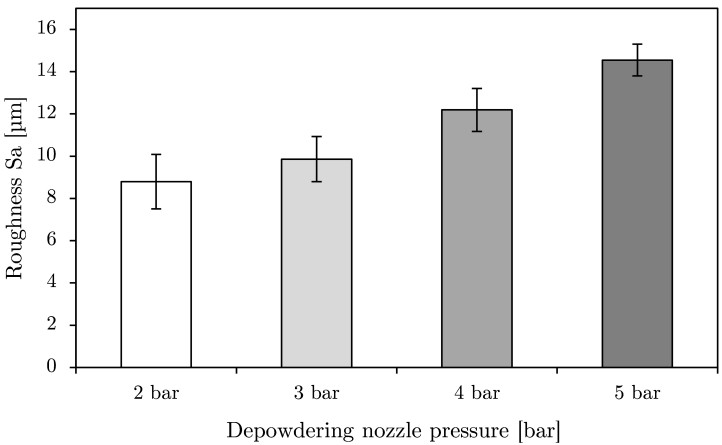

**Figure 11.** Surface characteristics for different depowdering air pressures.

It is found that increasing air pressures lead to an increase in surface roughness. The lowest roughness with a Sa = 8.8 μm is obtained with the depowdering pressure of 2 bar. At this low pressure, however, the depowdering takes more time and loose powder residues remain adhered to the component. These can then lead to contamination of the sintering furnace or the measuring equipment. Increasing the pressure to 3 bar leads to an increase in surface roughness to an average of Sa = 9.9 μm, and reliably removes the adhering powder in a sufficiently short time. This trend continues with further increases in the pressure to 4 or 5 bar. The highest level of roughness is achieved at 5 bar with a Sa = 14.6 μm, which means an increase of approximately 65% compared to the pressure level of 2 bar. The standard deviation decreases with increasing pressure. Whilst at 2 bar it is still approximately 1.3 μm, it decreases successively with each next higher pressure level until at 5 bar it is only 0.8 μm. The coefficient of variation decreases successively from 0.15 to 0.11 to 0.8 to 0.5. Nevertheless, the rather high standard deviations show that the manual depowdering process has only moderately good repeatability, even under strictly controlled process parameters.

### 3.2. Influence of the Print Orientation on the Surface Roughness

The results of the roughness analysis as a matter of printing orientation are shown in Figure 12. In accordance with the literature [3,6,29], it is shown in Figure 12a that upskin areas in the X–Y plane have better surface quality than the side surfaces, with an average roughness of Sa ≈ 9.1 μm. This results in a roughness of Sa = 14.5 μm on the sidewards-facing surfaces. The bottom surfaces of the specimens show a roughness in the order of magnitude of the side surfaces. Although 30 measurements were made per orientation, the standard deviation is very high. For the top surfaces, this is about 2 μm, which corresponds to a coefficient of variation of about 0.22. The measurements of the side surfaces show a standard deviation of 2.4 μm (coefficient of variation: 0.17) and the bottom surfaces a standard deviation of 1.6 μm (coefficient of variation: 0.11). Although both the bottom-side and side-surface show comparable roughness in terms of Sa values, the Sz values tend to show that the valleys, and, respectively, the peaks, of the side-surfaces are deeper or higher than those of the bottom-sides (see Figure 12b). However, very large standard deviation can also be seen here, which is why this statement is subject to a fair degree of uncertainty.

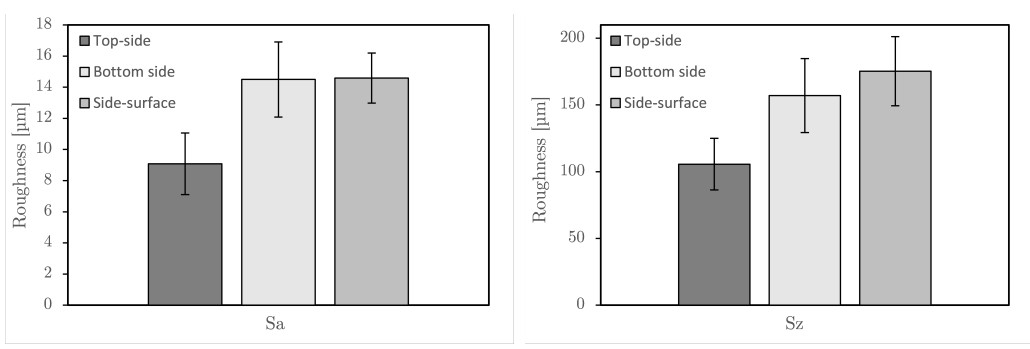

**Figure 12.** Surface roughness characteristics Sa and Sz for the three main printing orientations.

### 3.3. Influence of the Printing Orientation on the Sinter Joint

Images of the metallographically prepared sinter joints as a function of the print orientation are shown in Figure 13. It is immediately noticeable that the components have a high porosity arranged in lines, which is typical for the binder-jetting process. These lines run vertically in the middle image of Figure 13 because here the samples were printed at 90° to the build platform (referred to as "sidewards"). This so-called line porosity is the result of the layer-by-layer binder deposition process for bonding the discrete layers. After sintering, the components are firmly joined together. However, the joining zone is very easy to identify. Only very isolated joining points can be seen where the components have formed a bond during sintering. It is optically visible that the upwards-facing surfaces yield the highest contact ratio.

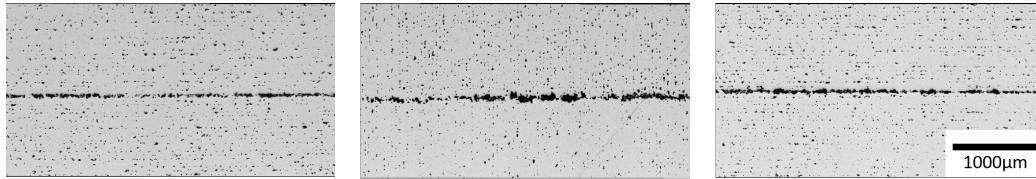

**Figure 13.** Formed sintered joint of two upper surfaces (**left**), two side surfaces (**middle**) and two bottom surfaces (**right**).

The analysis with Matlab shows a contact ratio of 13.5% on average for the top-side bonds, which is above that of the downwards-facing surfaces (11.5%) and significantly above the side-surface contacts (5.7%) (see Figure 14). Again, the evaluation shows a high standard deviation, which increases with increasing contact area ratio (absolute and relative). The gap height analysis shows the opposite behavior. The top-surface contacts show an average gap height of 29 μm, the bottom-surface contacts about 40 μm and the side-surface contacts 70 μm. The overall coefficient of variation is about 10% regardless of orientation. The analysis of the sintered joint generally shows that, irrespective of the component orientation, contact ratios are generally low. With this method, the bond strength may not be sufficient for all desired applications. Since the results show that the top surface contact is generally the highest, the following tests are performed with this setup.

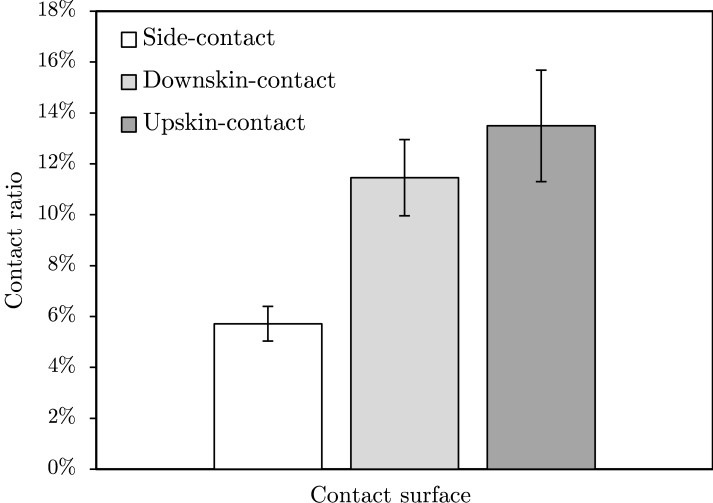

**Figure 14.** Contact ratio for different printing orientations.

### 3.4. Influence of Initial Contact Pressure on the Sinter Joint

The results shown in Figure 15 indicate that pressure application can increase the contact ratio of the sintered joint to about 22%, although there seems to be a degressive correlation between the applied force and the achievable contact ratio. Doubling the contact pressure from 50 to 100 N can increase the percentage of bond ratio by only 2% on average, whereas the increase compared to the contact ratio of the topsides without contact pressure is about 7%. In addition, the coefficients of variation are high, as in the previously described tests, and are approximately 0.16 for 0 N and 100 N contact force. Only the contact force of 50 N results in a lower coefficient of variation of 0.11. It must be noted that the force is not applied during sintering.

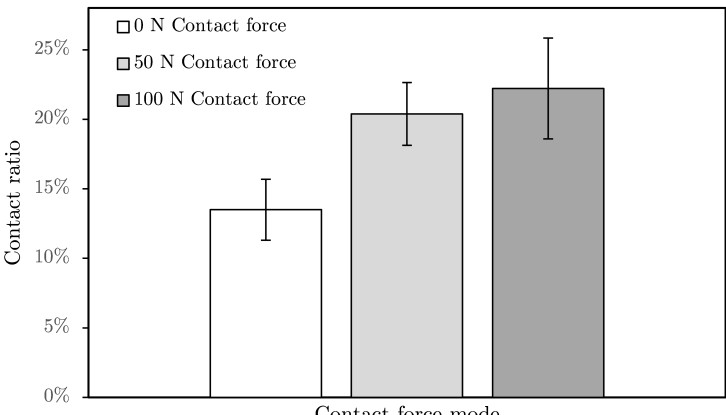

**Figure 15.** Contact ratio of the sintered joint as a function of the initial contact force.

### 3.5. Analysis of Simulation Results

The simulation was computed for different normal pressures and topographies. This section focuses on the influence of the topography on the contact area and the geometry of the gap. To compare the simulation results to the metallographically prepared sinter joints (compare Figure 13), a cross section of the simulation domain is shown in Figure 16. The simulation results are shown for a normal pressure of $100 \times 10^3$ Pa which corresponds to the 50 N initial contact force (compare Figure 15). The left image depicts the results for the upwards-facing surface and the right image shows the results for the sidewards-facing surface. The data plot aspect is identical for both images and the axes are evenly spaced. Please note the different exponents of the labels of the y-axes, which differ in one order of magnitude.

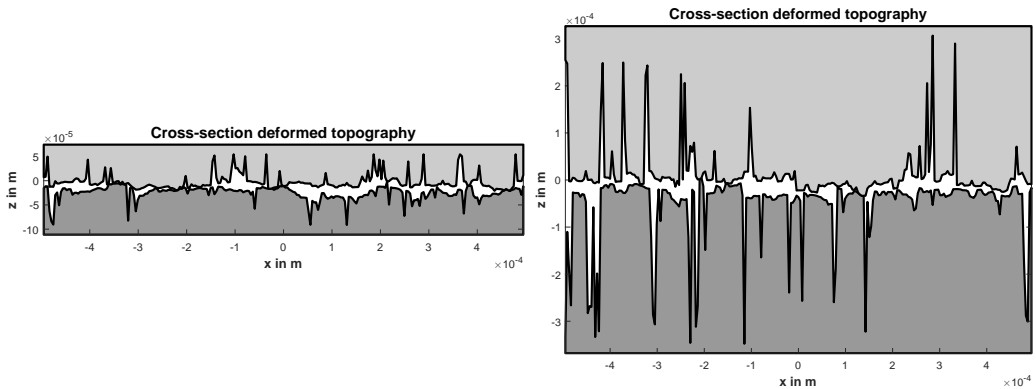

**Figure 16.** Cross-section of simulation results: (**left**) upwards- and (**right**) sidewards-facing surface.

The gap of the upwards-facing surface is much smoother than the sidewards-facing surface. This is a direct consequence of the original surface roughness, but this visualization emphasises the significance of the roughness. Furthermore, the contact points between the two topographies are visible. It is apparent that the gap geometry of the upwards-facing surfaces facilitates the formation of sinter bridges between molecules of the green parts, increasing the overall contact ratio.

To analyse the entire simulation domain with respect to its influence on the contact ratio, the distribution of the gap height was plotted in a histogram. The occurrence of specific gap height bins can then be compared between the two topographies. To depict contact cells, a bin of a separate color is used. Figure 17 shows the gap height histogram for the upwards-facing topography and Figure 18 for the sidewards-facing topography.

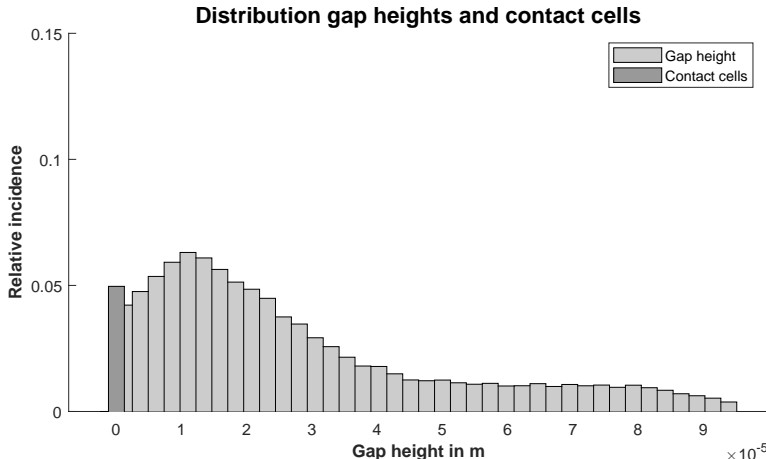

**Figure 17.** Distributions of deformed gap heights in the upwards-facing topography.

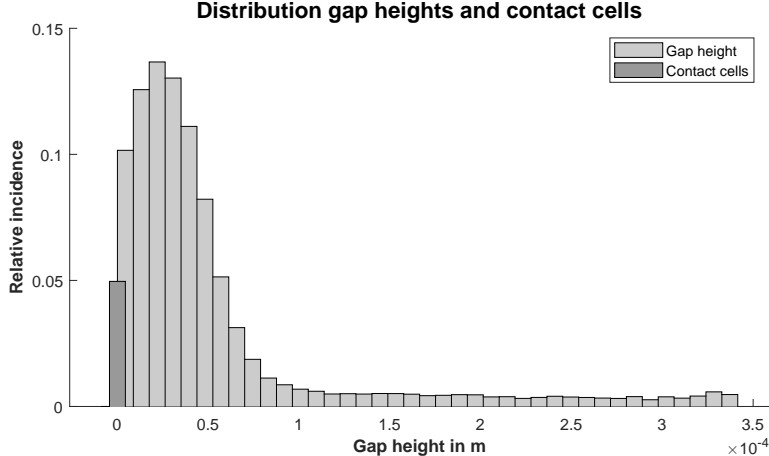

**Figure 18.** Distributions of deformed gap heights in the sidewards-facing topography.

The results show that the overall number of load-carrying cells, the contact ratio of the simulation, is similar for both topographies. This is expected, as the normal stress is limited due to the plasticity model, and thus if each contact cell transmits the stress limit the overall number will be roughly the same. The focus of the simulation, however, lies on the three-dimensional contact geometry. The histograms show the complex distribution of the local gap heights.

The histograms can be compared to the medium gap height that is measured in the experiment. In the experiments conducted for this study, the mean gap height for the upwards-facing topography is $29.7 \times 10^{-6}$ m. The mean gap height of the deformed simulation from Figure 17 is $30.2 \times 10^{-6}$ m, which is remarkably close. This height can be measured only after the sintering process, while the simulation represents the situation before sintering. Apparently, the sintering process does not significantly change the overall gap geometry but modifies cells that have a small gap height.

In the experiments conducted for this study, the mean gap height for the sidewards-facing topography is $70 \times 10^{-6}$ m. The mean gap height of the deformed simulation from Figure 18 is $61.7 \times 10^{-6}$ m. The comparison of the mean gap heights shows that the simulation is very close to the real gap geometry in the contact area.

### 3.6. Influence of Component Shape Deviation

It is found that the planarity of the specimen surfaces varies significantly. Figure 19 shows a cross-section with significant shape deviation (bent-up edges of the component). It is obvious that this reduces the bond strength and must be avoided. For the previous

sections, no specimens with significant shape deviations are evaluated and the deviated corners are cut from the evaluation.

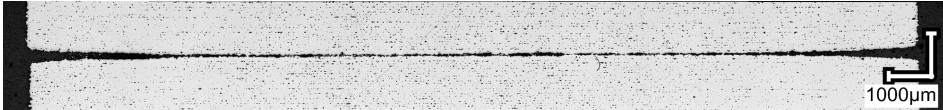

**Figure 19.** Sintered joint with macroscopic shape deviation.

It is found that the flatness of the components of a build job deviates on average by approximately 110 μm. The green parts have a positive deviation in the center and a negative deviation at the component edge (see Figure 20 right). The amount of the deviation depends strongly on the position of the component in the build chamber. The components positioned directly at the edge have a significantly better flatness of 89 μm than the components in the center, with an average deviation of 132 μm (see Figure 20 left).

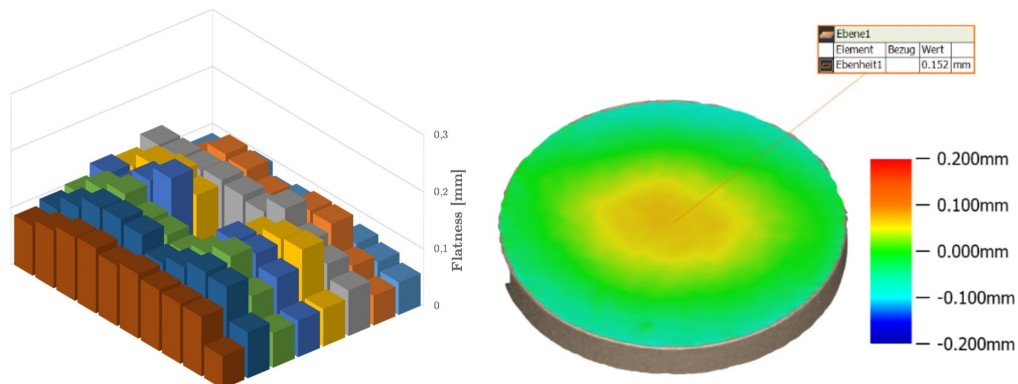

**Figure 20.** Flatness values distributed over the build area (**left**) and detailed image of the deformation of the components most likely due to the curing process (**right**).

## 4. Discussion

This work explores influences on the joint quality of sinter-bonded joints. The influence of the various production steps, such as print orientation, and parameters, such as depowdering pressure, can be broken down into influence on the surface roughness. The sintering process can compensate rough surfaces to a small degree, depending on the sinter parameters, but this effect is limited and smooth surfaces are key for an effective bond.

Depowdering of the green part cannot be avoided but leads to an increase in surface roughness. Since the absolute roughness values increase with an increase in pressure, it can be assumed that due to the insufficient strength of the green parts, the pressured air removes isolated powder grains or agglomerates bonded by the binder from the green part, thus increasing the roughness. Although a lower pressure achieves a lower roughness, depowdering takes slightly longer. Additionally, a longer manual depowdering process could lead to more variation in surface properties. At low pressures, it can be assumed that manual cleaning affects repeatability more than it does at higher pressures. Due to the manual execution, both time exposure in the compressed air jet and distance of the compressed air nozzle to the component surface may vary, and are a reason for the deviation in the results.

Overall, the air pressure of 3 *bar* was shown to be the best compromise between quality of powder removal and surface roughness. In addition to the selected pressure, the amount of powder present as dust in the air was also determined to be decisive. This occurs when the ratio between the vacuum power and the secondary air within the depowdering glovebox is not matched, or while depowdering internal structures. Existing powder dust is then blown onto the component with the compressed air lance and leads to erosive removal of surface particles. When dust load in the air is high, a trace of the compressed air lance

movement can even be seen on the component. This relationship should be investigated in further research, but was not the subject of the present study.

As already described, the influence of the print orientation on the roughness is known in the literature. The roughness of the sidewards-facing surfaces is higher, since here the stacked layers form the surface. This results in a surface with a roughness that is defined by the peaks and valleys of the powder layers. This strongly influences the surface roughness orthogonal to the layer build-up direction [3,29,36]. On the downwards-facing component surface, the applied binder penetrates the powder bed beyond the component contour to a small extent, so that a slight powder adhesion of the underlying layer leads to an increased roughness of the same order of magnitude as for the side surfaces [37–39]. This so-called bleeding effect could be enhanced by the special curing process of solvent extraction. The roughness of the top surface of the component is only affected by the impingement of the binder droplets when the recoating parameters are properly set, and is therefore smaller [40]. It is surprising that the standard deviations are very high, despite a very large number of samples. This indicates a general process instability and underlines the large influence of the manual depowdering process on surface quality.

Although the smoother top surface of the components bonded significantly better in the sintering process relative to the other orientations, not even 20% of the surfaces could be joined. Despite the fact that the roughness of the side faces and the bottom faces are equal in terms of magnitude, the side faces achieve a contact percentage that is just half that of the bottom faces. The gap height analysis shows that the surface pattern of the side faces due to the stacked print layers results in the average gap height of the side-face contacts being almost twice as high as that of the bottom face contact, which explains the significantly lower contact ratio. The use of the side surface as a contact surface has another major disadvantage. Due to inhomogeneous component shrinkage in the binder-jetting process, the green parts have to be positioned in such a way that the z-axes from the printing process are aligned. Otherwise, the components do not shrink in the same direction, which causes a serious misfit and induces stresses that in turn have a negative effect on the joining result. Since the shrinkage in the printing plane is almost homogeneous, this disadvantage does not affect the top or bottom sides.

The use of an initial contact force leads to a slight improvement in the contact ratio. The linear increase in the applied force does not show a linear response related to the contact ratio after sintering. Rather, the contact ratio approaches a maximum. The authors hypothesize that the general usage of a low contact force breaks local roughness peaks and thus increases contact, but beyond this, increasing the pressure does not yield a significant improvement.

The simulation gives insights into the formation of the gap geometry that is formed by elastic and plastic deformation caused by an initial contact pressure. This contact pressure can occur deliberately or as a result of the production process of the green parts. The distribution of the gap height influences the sintering process, for not only do the contact cells form the sinter bridges between the two green parts, but the cells with a small gap height are relevant for the bond as well.

During sintering, cells with a small gap height may form additional bonds between the two parts, strengthening the overall bond strength. It is also known from the MIM sector that further contacts can be created by so-called rolling and tilting processes of the powder particles during sintering [41]. It is found that for the upwards-facing surface, the amount of cells with a small gap height is much greater than the amount of small cells of the sidewards surface (compare Figure 17). This phenomenon might be responsible for the difference between the contact ratio of the simulation and the corresponding experiment (sinter joint after initial contact pressure). This leads to the conclusion that the contact ratio for the smoother surface is greater. It is concluded that the surface roughness is the dominant factor when it comes to comparing contact ratio.

The deviation in planarity of the specimen surfaces was tracked down to the special curing process of the protoype printers, but can also occur in other devices due to a variety

of reasons. Due to the differences in deformation between the edge area and the center of the build area, it is assumed that the generation of negative pressure for the extraction of the evaporated solvent through openings in the build platform is the cause of the deformations, since there are no extraction openings in the edge area on the build platform. The powder areas where binder has been introduced have a lower density than the surrounding powder bed (green part density $\approx 4.5\,\mathrm{g/cm^3}$, powder bed density $\approx 4.7\,\mathrm{g/cm^3}$). Although the density of the green parts is lower than that of the surrounding powder bed, the authors assume that the introduced binder closes the pore network of the powder. Thus, the air can be drawn more easily through the non-wetted areas, which could lead to the free powder compacting more strongly than the bound powder in the outer edges of the parts when negative pressure is applied. In the edge areas of the part, this could result in a shear force that warps the parts, which are still uncured and easily deformable at this point. Since there are no extraction openings in the edge area of the build platform, the loose powder surrounding the green parts may not be compacted as much here as in the middle of the platform. As a result, the deformation in the edge area occurs to a lesser extent. By placing two deformed surfaces on top of each other, the deviation adds up, so that any contact is prevented by the resulting gap. Although the cantilevered edge areas of the component halves sink back onto each other in the sintering process due to gravity, this is not sufficient to close the resulting gaps, even in the case of the less warped specimens.

## 5. Conclusions and Outlook

Complex parts from the binder-jetting process cannot be fully depowdered due to the insufficient green-part strength. This study proposes a new approach to print such components in two parts and to join them through material diffusion in the sintering process by simply placing them on top of each other. Achieving the printability of such shapes would enable the possible economical usage of binder jetting in many areas. For this purpose, different influencing parameters on the joint quality were analyzed and a new method was developed to evaluate the contact ratio metallographically. In addition, a simulation model was developed to determine the contact between green parts. The main conclusions are summarized as follows:

1. The air pressure during depowdering has a significant impact on the surface roughness of green parts. The lowest roughness of Sa $\approx 8.8\,\mu$m was measured at 2 bar. The roughness increases with increasing air pressure: 3 bar: Sa $\approx 9.9\,\mu$m; 4 bar: Sa $\approx 12.2\,\mu$m; 5 bar: Sa $\approx 14.6\,\mu$m.
2. Depending on the print orientation, the roughness increases: upskin areas in the X–Y plane have better surface quality than the side surfaces with an average roughness of Sa $\approx 9.1\,\mu$m. Downskin faces and side surfaces show the same, higher roughness of Sa $\approx 14.5\,\mu$m.
3. Despite 30 measurements per orientation, the roughness measurements of the surfaces show a large coefficient of variation (top surface: 0.22; side surface: 0.17; bottom surface: 0.11), which suggests that surfaces are currently poorly reproducible in binder jetting.
4. The main influence on the contact ratio is roughness. Therefore, the smoothest part surfaces, the top sides, achieved the relatively highest (nevertheless low) contact ratios during sinter joining (average 13.5%). Even though the resulting connection holds the parts in place, it does not create a mechanically stable joint.
5. An initial contact pressure (100 N) on green parts improves the contact ratio to about 22%. This might be sufficient for a mechanically unstressed component, but certainly not enough to create a fluid-tight connection. However, this would be particularly necessary for components with internal channels.

Even if the carried out studies show that it is possible to join binder-jetting components using the sintering process, the resulting joint achieves neither sufficient strength nor fluid tightness. To optimize the process, loose metal powder could be introduced into the joining zone to increase the contact ratio sufficiently. Solving the problem of depowdering

internal structures in binder jetting opens up a very large field of currently impossible-to-manufacture part shapes, which strengthens the economic use of the process.

**Author Contributions:** A.R.: Data curation, Formal analysis, Investigation, Methodology, Validation, Visualization, Roles/Writing—original draft, Writing—review and editing; L.S.: Data curation, Investigation, Software, Validation, Visualization, Roles/Writing—original draft, Writing—review and editing; M.M.: Conceptualization, Funding acquisition, Resources, Supervision, Validation, Writing—review and editing; F.P.: Project administration, Resources, Writing—review and editing; All authors have read and agreed to the published version of the manuscript.

**Funding:** We acknowledge support by the Open Access Publication Funds of Technische Universität Braunschweig.

**Data Availability Statement:** The raw/processed data required to reproduce these findings cannot be shared at this time as the data also form part of an ongoing study.

**Conflicts of Interest:** The authors declare no conflicts of interest.

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
