# Peer review of "Investigation and Simulation of the Surface Contact Characteristics of Sinter-Joined Binder Jetting Components"

_applsci, doi:10.3390/app12073478_

Round 1

Reviewer 1 Report

The article is devoted to the influence of various parameters that affect the strength of the connection and the contact surface between the two parts of the desired component. The study's relevance is dictated by Binder Jetting has great potential to revolutionize traditional manufacturing processes for high-performance components. However, today's applications face significant challenges in dedusting and cleaning complex internal geometries. A novel approach to solving these problems is to break the desired component into smaller, easily cleanable parts and use sintering to achieve the desired shape. This dramatically reduces manual cleaning and preparation time during production since the sintering process is necessary anyway. However, at present, the agglomerated bond is much weaker and may pose a risk of component failure. The study identifies and quantifies many influences in various experimental setups: dedusting air pressure, the orientation of components in the printer build space, initial contact pressure between green parts, and macroscopic deformation of components. The modified Boussinesq contact model confirms the experimental results. Combining experiments and simulations, the authors concluded that the close contact between the two green parts varies between 5-26% depending on the set of parameters used. The idea of two-component production by spraying a metal binder and subsequent connection of the components in the sintering process is presented.

Despite the satisfactory quality of the article, some shortcomings need to be corrected.

  1. English should be proofread.
  2. The aim of the article should be defined.
  3. The state-of-the-art methods should be separated from the ones proposed by the authors.
  4. Figure 5 should be described more in the text.
  5. The conclusion section should contain numerical results obtained by the authors.
  6. The scientific and practical novelty of the paper should be highlighted.

I recommend that the manuscript be accepted after major revision, including English proofreading, in summarizing my comments. 

Reviewer 2 Report

The authors introduce the approach of two or more parts manufacturing by metal binder jetting method and subsequent joining of the components in the sintering process. To implement this approach, the authors investigated surfaces characteristics of joined binder jetting components.  

It is important to note that the authors used relatively simple methods for evaluating unique surface characteristics. However, the results allowed to completely characterize specimens. A special thanks to the authors for a new methodological approach for evaluating joint quality (section 2.3).

Comments and Suggestions:

Section 2.1.

In this section should be added more information about the composition of the powder and the binder. 

Lines 34,35. Sentence: Remaining powder forms a material bond with the component after the sintering process significantly altering the desired component shape and function.

What does the phrase "material bond" mean? 

Section 3.1 in Results.

The authors used four values of air pressures for depowdering. However, time of depowdering is also different. In this section should be added more information about depowdering time for each air pressure and the influence of depowdering time on the roughness.

Lines 486-491 in the Discussion

The authors report about plastic deformation of specimens caused by an initial contact pressure. For conclusions about of type deformation the authors should be added the information about material  or rephrase this sentence.

Section 3.6.

How can the Authors explain the deviation of the flatness of the components distributed over the build area (figure 18 left)?

Reviewer 3 Report

The authors presented an article “Investigation and Simulation of the Surface Contact Characteristics of Sinter Joined Binder Jetting Components”. In the article the binder jetting technology (3D printing process for plastics and metals) and influences on the joint quality of sinter bonded joints was presented. It was shown that the roughness of binder jetting components is strongly influenced by the current manual depowdering process. The authors introduced the idea of two-part manufacturing in metal binder jetting and subsequent joining of the components in the sintering process The experimental results were supported by a modified Boussinesq contact model. The article can be interesting from an engineering point of view, however there are a few points in the article that require further explanation.

Comment 1:

Abstract

Demonstrate in the abstract novelty, practical significance. Add quantitative and qualitative work results to the abstract. Briefly describe the methods used in the research.

Comment 2:

  1. Introduction

There are several technical mistakes:

Line 38: dot should be in the end of the sentence (…function [5,6].)

Line 39: describe the abbreviation (…LPBF (Laser Powder Bed Fusion) process,…)

Line 46: add a space (…figure 1).

Line 51: dot should be in the end of the sentence (…geometry [7,8].)

Line 93: dot should be in the end of the sentence (…material [17].)

Line 98: dot should be in the end of the sentence (…effects [9].)

Line 108: dot should be in the end of the sentence (…strength [21].)

Line 112: dot should be in the end of the sentence (…the joint [23].)

Comment 3:

  1. Methods

This chapter should be more elaborate. Describe in more detail the equipment used for 3D printing and the sintering of the specimens. Add if possibly the pictures of the machines and the measurement apparatus.

Authors use

Comment 4:

  1. Conclusions and outlook

Add quantitative and qualitative work results. In addition, it is necessary to more clearly show the novelty of the article and the advantages of the proposed method. What is the difference from previous work in this area? Show practical relevance. Presented conclusions are only a description of the test results. Conclusions should reflect the purpose of the article.

Comment 5:

Article is interesting. The literature is very well chosen. After minor changes the reviewed article can be considered for publication in the Applied Sciences journal.

Round 2

Reviewer 1 Report

Thanks for the authors for considering the reviewer's comments and recommendations. In my opinion, now the article can be accepted.